# Applying the Action Principle of Classical Mechanics to the Thermodynamics of the Troposphere

Ivan R. Kennedy [1,*] and Migdat Hodzic [2]

1    Institute of Agriculture, School of Life and Environmental Sciences, University of Sydney, Sydney, NSW 2006, Australia

2    Faculty of Information Technologies, University Dzemal Bijedic in Mostar, 88000 Mostar, Bosnia and Herzegovina; migdat.hodzic@edu.fit.ba

*    Correspondence: ivan.kennedy@sydney.edu.au; Tel.: +61-413071796

**Abstract:** Advances in applied mechanics have facilitated a better understanding of the recycling of heat and work in the troposphere. This goal is important to meet practical needs for better management of climate science. Achieving this objective may require the application of quantum principles in action mechanics, recently employed to analyze the reversible thermodynamics of Carnot's heat engine cycle. The testable proposals suggested here seek to solve several problems including (i) the phenomena of decreasing temperature and molecular entropy but increasing Gibbs energy with altitude in the troposphere; (ii) a reversible system storing thermal energy to drive vortical wind flow in anticyclones while frictionally warming the Earth's surface by heat release from turbulence; (iii) vortical generation of electrical power from translational momentum in airflow in wind farms; and (iv) vortical energy in the destructive power of tropical cyclones. The scalar property of molecular action ($@_t \equiv \int mvds$, J-sec) is used to show how equilibrium temperatures are achieved from statistical equality of mechanical torques ($mv^2$ or $mr^2\omega^2$); these are exerted by Gibbs field quanta for each kind of gas phase molecule as rates of translational action ($d@_t/dt \equiv \int mr^2\omega d\phi/dt \equiv mv^2$). These torques result from the impulsive density of resonant quantum or Gibbs fields with molecules, configuring the trajectories of gas molecules while balancing molecular pressure against the density of field energy ($J/m^3$). Gibbs energy fields contain no resonant quanta at zero Kelvin, with this chemical potential diminishing in magnitude as the translational action of vapor molecules and quantum field energy content increases with temperature. These cases distinguish symmetrically between causal fields of impulsive quanta ($\Sigma h\nu$) that energize the action of matter and the resultant kinetic torques of molecular mechanics ($mv^2$). The quanta of these different fields display mean wavelengths from $10^{-4}$ m to $10^{12}$ m, with radial mechanical advantages many orders of magnitude greater than the corresponding translational actions, though with mean quantum frequencies ($\nu$) similar to those of radial Brownian movement for independent particles ($\omega$). Widespread neglect of the Gibbs field energy component of natural systems may be preventing advances in tropospheric mechanics. A better understanding of these vortical Gibbs energy fields as thermodynamically reversible reservoirs for heat can help optimize work processes on Earth, delaying the achievement of maximum entropy production from short-wave solar radiation being converted to outgoing long-wave radiation to space. This understanding may improve strategies for management of global changes in climate.

**Keywords:** quantum of action; action mechanics; vortical action; principle of least action; quanta; Gibbs energy; Gibbs field; Carnot cycle; wind turbines; downwelling radiation; tropical cyclone; gravitational energy; climate science



## 1. Introduction

The primary purpose of this article is to demonstrate how quantum fields sustain the action of molecular mechanics allowing heat energy to do internal work. An analysis of this linkage with our introduction of the action principle of classical mechanics [1,2] recognized

action as stated by Maupertuis and Euler, the integral of momentum ($mv$) with respect to the molecular trajectory ($\int mvds \equiv \int mr^2\omega d\phi \equiv mr^2\omega\delta\phi$, J.s, with $rd\phi$ or $ds$ spatial motion). In this form, action has the same dimensions as the vector state of angular momentum ($mr^2\omega$) since the differential for the phase angle ($d\phi \equiv \omega dt$) is dimensionless, signaling a dynamic change in spatial configuration. The action ($h$) of quanta at all wavelengths has the same value, a fundamental natural constant defined [3] by Planck ($6.62607 \times 10^{-34}$ J.s). Action of material particles is quantified as a function of radial curvature proportional to $1/r$ using radial coordinates; indeed, sustained rather than brief rectilinear motion is regarded as very unusual, except for the trajectories of quanta.

Previously, it was shown [1] that the total thermal energy required to heat all atmospheric gases from absolute zero to ambient temperatures is invariably several times their final kinetic energy. This is regarded as a starting point for establishing the thermodynamics of independent molecules sustained in air against gravity by quantum fields of energy; these fields are required to overcome various forms of binding, such as phase changes of melting from solid to liquid and liquid to vapor, as well as providing impetus for kinetic energy. Effectively, the motion of the individual molecules is mobilized by the Gibbs field energy. This total heat requirement is signified by the increase in entropy accurately calculated [1] from the action of each molecule's mechanical state. The extra heat requirement comprises the quantum fields sustaining the various internal degrees of freedom of Brownian action of the independent molecules of suspended gases.

Surprisingly, the more dilute a molecule is in the atmosphere, the greater the quantum field energy needed to provide its impetus; this is a conclusion that was made much earlier by Carnot in 1824 [4] in terms of the amount of *calorique* required in the working fluid of heat engines. These facts regarding complementary quantum fields and states of molecules have been largely neglected, given the widespread preference for a material description for molecules and their internal translational, rotational, and vibrational energy. The extra motion caused by heating has been considered self-sustaining from the principle of rectilinear inertia. Despite widespread opinion since Clausius' conclusion [2] that heat can disappear during work, it is proposed that quantum field energies are essential to sustain the torques exerted as molecular temperature supporting such work; such a thermodynamic force field was originally proposed in Einstein's theory [5] of Brownian movement of particles.

Our recent revision of the reversible heat engine cycle showed how Carnot's version of *calorique* could be seen as a surrogate for the Gibbs quantum field [2]. Carnot [4] distinguished between *chaleur* as heat available from burning coal and *calorique* for heat as a property of state of the working fluid required in the cylinder of a heat engine. He recognized that the difference between the quantity of *chaleur* absorbed isothermally by the working fluid from a hot source, such as a coal furnace (designated by Carnot as (*a*) and the lesser amount of *calorique* needing to be removed at the temperature of a much colder sink (*a'*), allowing the cycle to return to its initial state), represented the maximum work (*a–a'*) possible by the heat engine. Carnot was deprived of recognition for these findings because of his early death from cholera and the neglect of his manuscripts for 50 years. To better understand his inspiration, his cycle was examined in detail, reinterpreting caloric as consisting of a field of quanta. This article will show how our direct method [1,2] to estimate the absolute Gibbs energy from molecular data for gases establishes the configurational state of the working fluid. It is proposed that the statistical value of temperature for molecules can be understood dynamically as torque or a common rate of exchange of action if at equilibrium.

In this article, a novel linkage will be fashioned between quantum fields and molecules exhibiting action, using as examples a group of natural systems operating in our everyday world. To a large extent, previous observations regarding fields of quanta have not been pressed to logical conclusions, perhaps from too much reliance on partial differential equations to provide solutions to natural phenomena. Instead, a novel revision based on integrating the physical property of action [1,6] as a surrogate for molecular temperature

and density is recommended. Indeed, this common property of the momentum of faster quanta matching that of material particles may also provide a unifying principle for the mechanics of many scientific disciplines.

## 2. Basic Theory and Methods

In this section, some background regarding the principle of least action and its significance in classical mechanics will be given. Equations for the estimation of entropy and energy in Gibbs fields will also be introduced.

### 2.1. Complementary Fields for Virtual Quanta and Dynamics of Material Particles

Our investigations of quantum action fields and interactions between quanta and matter will be undertaken across large variations in scale, bridging, in part, the gap between quantum mechanics and gravity. These practical illustrations include the following topics, given as new forms of applied mechanics.

- We use data from our previous action revision of the Carnot cycle [2] to examine the thermodynamics of the working fluid in expressing power from heat engines in terms of variable quantum fields. Using Carnot's cycle, the study showed how the difference between the quantity of *calorique* obtained from a hot source and that absorbed by a colder sink gives the maximum power output possible from any heat engine while returning the engine to its initial thermodynamic state. The total energy (*U*) of the working fluid is not the source of the external work performed, but merely a scaffold for the variation in Gibbs quantum fields between the 2 distinct isothermal states that allow maximum work. A method to define the mean wavelength and frequency of the field quanta is given.
- By applying action mechanics to sustain the thermal and gravitational structure of the troposphere [7], we show how the virial theorem establishes the temperature gradient with altitude rather than adiabatic expansion. The quantum field characteristics of increasing altitude are calculated in this article, partitioned between translational and rotational energies.
- We illustrate the role of the vortical action ($\Sigma mR^2\omega$) of concerted molecular flow in anticyclones and cyclones as a higher degree of freedom storing thermal energy as vortical work, capable of warming the surface by turbulent friction.
- We estimate the maximum potential of vortical wind power in the Earth's atmosphere, already foreshadowed in an article now in review explaining a new method to calculate the maximum power of wind turbines [8].
- We explain the destructive power of tropical cyclones derived from solar heat consumed in the evaporation of seawater, which also uses vortical energy dependent on quantum impulses at the molecular level.

### 2.2. The Principle of Least Action Exemplified in Gibbs Field Mechanics

An appeal to the principle of least action is taken as justification for regarding molecular thermodynamics as action mechanics. Previous articles [1,2] abbreviated action as having the same physical dimensions as angular momentum ($J \equiv mr^2\omega$), stressing that it is a scalar function indicating change in configuration. It includes a spatial variation in the dimensionless property of angular motion ($@ \equiv mr^2\omega\delta\phi \equiv J\delta\phi$) by reference to a local couple and distant stars. Symmetry factors may also be needed to account for symmetrical pairing of particles or repeated structures. Any variation in action as the rate of change of this angle ($d\phi/dt = \omega$) can be expressed as a variation in torque ($mr^2\omega^2$).

For the Lagrangian (*L*) expressed as the difference between kinetic and potential energies ($L = T - V$) in a conservative system of constant total energy ($E = T + V$), the exact equivalence of our version of action (@) with the mathematics of variations version of

action ($S$) as the integral $\int_0^t L dt$ with respect to time is claimed. Both symbols refer to the same physical property.

$$S = \int_0^t (T - V)dt = \int_0^t (\frac{mr^2\omega^2}{2} - (-mr^2\omega^2))dt \tag{1}$$

$$= \int_0^t mr^2\omega^2 dt = \int_0^t J\omega dt = J\int_0^t \omega dt = J\int_0^t d\phi = J\delta\phi \tag{2}$$

In a conservative system without friction, any variation in kinetic energy ($T$) is matched by an opposite variation in potential energy ($V$) so that the total variation in the Lagrangian is proportional to the integral, with respect to time, of twice the kinetic energy ($2T$) as shown in Equation (2). Given that angular momentum ($mr^2\omega = J$) is conserved, the remaining variation ($\delta\phi$) indicates the change in phase angle $\phi$ in the time integrated ($t$). Therefore, the variation of the Lagrangian for a system with constant angular momentum can be represented entirely as the variation of angular motion with respect to time. The action per unit time, or torque ($mr^2\omega^2$), is maximal where the rate of change ($d\phi/dt = \omega$) is maximal. The minimum action per orbit must be $2\pi J$. Obviously, the action per radian must be stationary as $J$. For centers of force, such as molecular interactions or gravity, any deviation from an elliptical orbit will increase the pathlength of the Maupertuis action considered as $\int mvds = \int mr\omega.rd\phi = J\int d\phi = J\delta\phi$. Only in systems where the relative magnitude of quanta is very large will deviation of orbits as smooth curves be observable, such as for the motion of electrons.

Least action as given in the Feynman lectures [9] is expressed in terms of a Lagrangian consisting of the difference between kinetic energy and variation in gravitational potential energy ($mgh$). However, this relationship can be shown as equal to Equation (3) considering that gravitational acceleration ($g$) is not constant with height or radius from the center. Known to Newton, any parabolic trajectory on Earth is equivalent to a comet-like orbit towards its center, but with the fall intercepted by the Earth's surface.

Then the following integral of the Lagrangian as action ($S$) in Equation (3) can be applied, where $r_1$ and $r_2$ are radii to the Earth's gravitational center. Analogous processes are assumed to occur in many-bodied systems of molecules, with quantum fields generating the same torques and temperatures for each molecular species.

$$S = \int L dt = \int (\frac{mv^2}{2} - mgh)dt = \int (\frac{mv^2}{2} - m(r_1\omega_h^2 - r_2\omega_2^2)r)dt \tag{3}$$

Given that gravitational acceleration (ca. $r\omega^2$) decreases with height or radius, the Lagrangian (and its integral action) will include the same torque-like forms as in Equation (3), just as in Equation (1). This theoretical account is claimed as equally valid for many-bodied molecular systems in a microcanonical heat bath variable in temperature and torque, as for a gravitational system.

Care must be taken to distinguish potential energy (as given in Equation (3)) as orbits of constant relative action ($J$) from that in systems where the action varies because of increased angular momentum but decreased torque, as is the case with the electron in the hydrogen atom, for action in a higher gravitational orbit, or following a change in molecular temperature on heating. As a central force, a lower torque will be exerted corresponding to a lower field intensity for an electron on a longer radius, in accordance with the virial theorem. Absorption of quanta in the field causing a new state of longer radius will diminish the kinetic energy by an amount equal to the energy ($h\nu$) of the quantum, raising the potential energy by twice the value of the quantum or the decrease in kinetic energy. A future collapse of the new quantum state would involve emission of the quantum of energy plus an increase in kinetic energy of the same value as the emitted quantum. Showing this is a critical test in quantum mechanics [2]. The action of the orbit will vary according to the number of quanta involved, since all quanta have unit action, as was revealed by Planck [3].

The mathematics of the Lagrangian do not apply to such changes in state on absorption or emission of quanta from the field, since this orbital energy is not conserved.

### 2.3. Mathematical Basis and Procedure for Estimating Dynamic Action and Quantum Fields

For diatomic atmospheric gases such as $N_2$, the complex macroscopic variables of temperature and volume or density expressing entropy ($S$) and heat ($ST$) of statistical mechanics have been integrated in simplified form [1] as functions of translational action ($@_t = mrv$) and rotational action ($@_r = I\omega$), including factors for symmetry ($z_t$, $\sigma$). $z_t$ includes a term for the ratio of mean to root mean square velocity. This process involved a reinterpretation [1] of the partition functions for translation and rotation, given in most textbooks on statistical mechanics.

$$
\begin{aligned}
S_t T &= RT \ln[e^{5/2}(3kTI_t)^{3/2}/(\hbar^3 z_t\sigma)] & &\text{Translation} \\
S_t T &= RT \ln[\{8\pi^2(8\pi^3 I_A I_B I_C)^{1/2}(kT)^{3/2}\}/\sigma_r h^3] + 3/2RT & &\text{3 - dimensional rotation} \\
ST &= RT \ln[e^{9/2}@_t/\hbar^3 @_r/\hbar^2] & &\text{Translation + 2 - dimensional} \\
& & &\text{rotation}
\end{aligned}
\tag{4}
$$

$$
RT \ln[e^{9/2}n_t^3 j_r^2] \qquad\qquad \text{As quantum numbers} \tag{5}
$$

In these equations, the equivalence of macroscopic properties of temperature and pressure are shown as given by the $\log_e$ of action ratios ($@/\hbar$). Planck's reduced constant ($\hbar = h/2\pi$) is used in the denominator to measure the mean levels of the translational or rotational quantum states. Here, e indicates a mathematical exponent for internal energy and the subscripts refer to translation (t), rotation (r), and the 3 moments of rotational inertia (A, B, C) for 3-D molecules. This is classical statistical physics mechanics recast in a more holistic form. The heat capacity exponential term depends on whether the system volume is constant ($C_v$) or is able to expand at atmospheric pressure ($C_p$). In the following analysis, molecular pressure ($p$) for gases is expressed as a mean value for each species by the ideal gas law, shown in Equation (6), where N is the number of molecules per unit volume ($1/a^3$).

$$
p = kT/a^3 = NkT \tag{6}
$$

$kT$ is equivalent to root mean square velocity in $mv^2/3$, where $a$ is the mean cubic separation of molecules ($r = a/2$); each molecule is regarded as confined to its mean specific volume of $a^3$.

By modifying the Willard Gibbs statistical mechanics of extension in phases using action mechanics [1,2,6] it is possible to estimate absolute values for action and entropy, quantum state numbers per molecule, and the mean translational, rotational, and vibrational quanta in the field. Equation (7) gives a formula for estimating mean molecular properties for entropic energy per molecule as ($ST$), energy [$E$, $C_v T$, J], or enthalpy [$H$, $C_p T$, J] and Gibbs energy [$-G$, J], though using lower case as energy per molecule. It is easier to consider average values for molecules. In a variable pressure volume system undertaking reversible mechanical work such as the Carnot cycle, if estimating the molecular energy ($e$), $c_p T$ must be replaced by $c_v T$ with $c_v$ equal to 3/2 for a monatomic gas such as argon. No work is undertaken against the atmosphere, with the internal field energy varying with internal pressure reversibly, with respect to the back pressure from the external work. For maximum efficiency, the whole cycle must be performed reversibly, as Carnot defined.

$$
s_t T = c_v T + kT\ln[(@_t/\hbar)^3] = c_v T + 3kT\ln[(n_t)] = e - g_t \tag{7}
$$

The Gibbs energy per molecule ($g_t$) is 0 at absolute 0, where the temperature is 0 K and becomes progressively more negative as the temperature increases, and the Gibbs field spontaneously gains quantum energy. This is consistent with the Maxwell relationship where $\Delta G$ equals $\Delta H - T\Delta S$ and spontaneous reactions have negative changes in Gibbs

energy, as is the case when entropy increases. In Equation (7), kT is equal to $mv^2/3$ for translational kinetic energy.

In principle, the same procedure used previously [1,2,6] is applied throughout this article. This is based on achieving a field condition where the time-integrated momentum exchange from impulses caused by the quantum field ($\Sigma h v_i/c$) and the reversible momentum exchange for the material particles ($\Sigma m v_i$) is equal within the same Brownian [5] or random walk matrix. Per se, neither material particles nor quanta exchange momentum directly and particles only do so in collisions by way of the far swifter intervening quanta. In terms of the rate of action impulses, establishing the number of quanta keeping the material particles separated is effectively magnified by the factor $c/v_i$; so, a simple comparison of the ratio of momentum, either ($h v_i/cmv_i$) or ($h/\lambda_i m v_i$), will be greater by this factor than is needed for quanta in the field. This hypothesis will be tested during the various applied exercises as follows.

1.  The mean translational (or rotational) action ($@_t = mrv_t$) of molecules is estimated for the molecular field based on macroscopic concentration and temperature, including any effect of symmetry that multiplies the probability of field energy interacting with particular groups, reducing free paths. The more symmetry exhibited in a mechanical system, the lower the action and the field energy needed to sustain the system [1]. To estimate translational action ($@_t = mrv_t$), the mean velocity is required rather than the root mean square velocity, which is approximately 1.09 times less. The simple methods used in action mechanics to calculate entropy and absolute Gibbs energy based on molecular properties were applied in a paper [6] examining thermodynamics of $H_2$ and its lysis to hydrogen atoms at the temperature of the sun's surface. Its difficult formation by thermal decomposition of water above 4500 K and by a much easier reversible formation from ammonia in the Haber process near 400 K were shown.
2.  The mean number ($n_t$) for translational quantum microstates per molecule for current mechanics is extracted by the ratio of the mean molecular action to Planck's reduced quantum of action ($n = @/\hbar$). For all cases examined in this article, this ratio exceeds unity by a significant margin, indicating a high entropy for this degree of freedom. For this reason, these translational processes all behave classically given the low rate of occupancy of quantum microstates.
3.  The absolute value of the translational Gibbs energy ($g_t$) is then estimated as a logarithmic function of the number of quantum microstates, as published previously [1,2,6]. As this value becomes more negative, the field quantum energy increases, as anticipated by the second law of thermodynamics.
4.  The mean value of virtual quanta in the field is then calculated ($hv = -g_t/n_t$), enabling the virtual frequency and wavelength in the field to be estimated. Peak values for translational quanta will reflect the vis viva, twice the kinetic energy for the Carnot cycle ($mv^2 = 3kT$). For other processes, such as the dynamics of air molecules in wind, the vis viva involved is very low and according to wind speed, indicating a very low temperature using the 1-D relationship $mv^2 = kT$.

It is assumed in this theory that no quantum coherence between molecular quantum states operates, as would occur if kinetic temperature was too low to ensure significant occupation of quantum states. This action-based method focuses on translational dynamics of particles for reversible heat and work. Given the very small size of translational quanta at ambient temperatures, all processes dealt with here approximate classical physics.

## 3. Results and Discussion

In this study, previous research is described showing how molecular systems must have complementary fields of quantum energy specific to each species of molecule. This will demonstrate specific thermodynamic properties varying with temperature and pressure expressed more holistically as molecular action states, supported by the quantum field action designated as the Gibbs field.

### 3.1. Revising the Carnot Cycle as a Basis for a Gibbs Action Field

Equation (7) shows that the total heat content needed is the sum of the internal energy ($c_v T$) plus the absolute energy of the field ($-g_t$), which becomes more positive as its quantum state increases. Removing temperature ($T$) in Equation (7) gives the absolute entropy per molecule of the quantum state under the current environmental conditions of temperature and pressure. Table 1 reproduces and extends data from our earlier article [2]; shown are relationships between matter and quantum fields relevant to all four stages proposed by Carnot as reversible, determining the most efficient generation of power in the heat engine cycle. The table shows that Carnot's formal explanation of the cycle using caloric is consistent with quantum theory, with its modern surrogate shown as mean negative Gibbs energy per molecule ($-g_t$). Carnot specifically indicated that the maximum possible work was equal to the second differences of *a*–*a'* or *b'*–*b*, where *a*, *b*, *a'*, and *b* were primary differences between absolute Gibbs energy values calculated for argon and nitrogen shown in Table 1. For two working materials as ideal gases, the following conclusions from the heat engine cycle are made, considering the impulsive quantum properties of the working fluid as causal. Most formulae, such as the Schrödinger wave equation, estimate quanta absorbed or emitted as the difference between states, but Table 1 gives their absolute mean values. Note that these values for translational action ($@_t$) are corrected here for a simple programming error in reference [2] that underestimated action by a factor of 1.47.

- At all four stages of the cycle, the relative action ($@$) of the working fluid calculated indicates its entropy state according to Equation (7). Gibbs energy ($G_t$ or $g_t$) is always zero or negative, decreasing from minimum action near absolute zero K. Uniquely, action mechanics quantifies the Gibbs field here as mean numbers of virtual quanta needed per molecule to sustain their temperature and pressure.
- Atmospheric pressure is not relevant to the enclosed Carnot cycle, so from Equation (7) all effects of changes in pressure in the cycle can be calculated as changes in Gibbs energy calculated from macroscopic temperature and pressure, given these are equivalent [2,6]. Shown in Table 1, the field of virtual quanta ($\Sigma h\nu$) contains almost 10 times as much field energy (largely provided in melting and vaporization) as the kinetic energy of the material particles, sustaining molecular torques ($mv^2$) and material pressures.
- Each turn of the Carnot cycle shown in Table 1 is assumed to absorb $kT$ of heat from the hot source of quanta appropriate for the temperature and pressure and the same quantity $kT$ removed at the colder sink as different quanta of lower frequency.
- The pressure values shown in the table also produce the ratio of torque intensity per unit volume ($mv^2/3a^3$ or $kT/a^3$) to the negative Gibbs energy density ($-g_t/a^3$) or mean density of virtual quanta held within the mean volume $a^3$ occupied by each molecule. For argon, this energy ratio is constant for transitions in adiabatic or isentropic states with no change in heat content. Where isothermal processes at constant temperature (or torque) occur, there is a change in this ratio as heat is added or removed.
- For nitrogen, the interaction between quantum cells for translation and rotation requires that the product of the quantum densities, shown as ($n_t^3 \times j_r^2$), respectively, in the table, must remain constant for adiabatic processes that are isentropic, exhibiting constant action.

**Table 1.** Action thermodynamics of the Carnot cycle for working fluids of argon and nitrogen molecules.

| Property | Stage 1 | Stage 2 | Stage 3 | Stage 4 |
|---|---|---|---|---|
| Kelvin temperature | 640–640 | 640–288 | 288–288 | 288–640 |
| Argon (Ar) | Isothermal | Isentropic | Isothermal | Isentropic |
| Radius ($a/2 = r$, m) | $6.410895 \times 10^{-10}$ | $8.947125 \times 10^{-10}$ | $13.337586 \times 10^{-10}$ | $9.556798 \times 10^{-10}$ |
| Pressure ($kT/a^3$, J/m$^3$) | $4.191891 \times 10^6$ | $1.542111 \times 10^6$ | $0.2094820 \times 10^6$ | $0.5694312 \times 10^6$ |
| Translational action (@$_t$, J.s) | $12.43697 \times 10^{-33}$ | $17.35719 \times 10^{-33}$ | $17.35719 \times 10^{-33}$ | $12.43697 \times 10^{-33}$ |
| Mean quantum number ($n_t =$ @$_t/$) | 117.932 | 164.587 | 164.587 | 117.932 |
| Negative Gibbs energy ($-g_t$, J) | $12.6446 \times 10^{-20}$ (a) | $13.5282 \times 10^{-20}$ (b') | $6.0877 \times 10^{-20}$ (a') | $5.6901 \times 10^{-20}$ (b) |
| Mean quantum ($h\nu$, J) | $1.07220 \times 10^{-21}$ | $0.82195 \times 10^{-21}$ | $0.36988 \times 10^{-21}$ | $0.48249 \times 10^{-21}$ |
| Energy density ($g_t/a^3$, J/m$^3$) | $5.998728 \times 10^7$ | $2.361020 \times 10^7$ | $0.320724 \times 10^7$ | $0.814874 \times 10^7$ |
| Quantum frequency ($\nu$, Hz) | $1.61812 \times 10^{12}$ | $1.24045 \times 10^{12}$ | $0.55820 \times 10^{12}$ | $0.72815 \times 10^{12}$ |
| Wavelength (m) | $1.85272 \times 10^{-4}$ | $2.41680 \times 10^{-4}$ | $5.37066 \times 10^{-4}$ | $4.11716 \times 10^{-4}$ |
| $\lambda/2\pi r$ (quanta/molecular) | $4.59951 \times 10^4$ | $4.29909 \times 10^4$ | $6.40870 \times 10^4$ | $6.85654 \times 10^4$ |
| Molecular frequency ($\omega$) | $9.81843 \times 10^{11}$ | $7.03521 \times 10^{11}$ | $3.16585 \times 10^{11}$ | $4.41829 \times 10^{11}$ |
| Ratio ($\nu/\omega$) | 1.64804 | 1.76321 | 1.76321 | 1.64804 |
| Pressure ratio ($g_t/kT$) | 14.3103 | 15.3103 | 15.3103 | 14.3103 |
| $nh/\lambda mv \times 10^{-5}$ | 1.0024 $c/v = 4.77 \times 10^5$ | 1.0732 $c/v = 4.77 \times 10^5$ | 1.0732 $c/v = 7.1 \times 10^5$ | 1.0024 $c/v = 7.1 \times 10^5$ |
| Nitrogen (N$_2$) translational | | | | |
| Radius ($a/2= r$, m) | $6.410895 \times 10^{-10}$ | $8.947125 \times 10^{-10}$ | $17.40496 \times 10^{-10}$ | $12.47120 \times 10^{-10}$ |
| Pressure ($kT/a^3$, J/m$^3$) | $4.191891 \times 10^6$ | $1.542111 \times 10^6$ | $0.942669 \times 10^5$ | $2.56244 \times 10^5$ |
| Translational action (@$_t$, J.s) | $10.40552 \times 10^{-33}$ | $14.52207 \times 10^{-33}$ | $18.95066 \times 10^{-33}$ | $13.5787 \times 10^{-33}$ |
| Mean quantum number ($n_t$) | 98.669 | 137.703 | 179.697 | 128.758 |
| Negative Gibbs energy ($-g_t$, J) | $12.1719 \times 10^{-20}$ | $13.0555 \times 10^{-20}$ | $6.1925 \times 10^{-20}$ | $5.79484 \times 10^{-20}$ |
| Mean quantum ($h\nu$, J) | $1.23361 \times 10^{-21}$ (a) | $0.94809 \times 10^{-21}$ (b') | $0.34461 \times 10^{-21}$ (a') | $0.45006 \times 10^{-21}$ (b) |
| Energy density ($g_t/a^3$, J/m$^3$) | $5.774446 \times 10^7$ | $2.278515 \times 10^7$ | $0.146810 \times 10^7$ | $0.37345 \times 10^7$ |
| Quantum frequency ($\nu$, Hz) | $1.86172 \times 10^{12}$ | $1.43082 \times 10^{12}$ | $0.52007 \times 10^{12}$ | $0.67921 \times 10^{12}$ |
| Wavelength (m) | $1.61030 \times 10^{-4}$ | $2.09525 \times 10^{-4}$ | $5.76450 \times 10^{-4}$ | $4.41385 \times 10^{-4}$ |
| $\lambda/2\pi r$ (quanta/molecular) | $3.99768 \times 10^4$ | $3.72712 \times 10^4$ | $5.27119 \times 10^4$ | $5.04682 \times 10^4$ |
| Molecular frequency ($\omega$) | $11.73527 \times 10^{11}$ | $8.40869 \times 10^{11}$ | $2.89965 \times 10^{11}$ | $4.04678 \times 10^{11}$ |
| Ratio ($\nu/\omega$) | 1.58664 | 1.70159 | 1.79355 | 1.67839 |
| Pressure ratio ($g_t/kT$) | 13.77530 | 14.77530 | 15.57381 | 14.57381 |
| $nh/\lambda mv \times 10^{-5}$ | 1.1541 $c/v= 3.99 \times 10^5$ | 1.2379 $c/v= 3.99 \times 10^5$ | 0.8753 $c/v = 1.14 \times 10^5$ | 0.8191 $c/v = 1.1 \times 10^5$ |
| Nitrogen rotational | | | | |
| Negative rotational Gibbs energy ($-g_r$, J) | $4.1575 \times 10^{-20}$ (a) | $4.1575 \times 10^{-20}$ (b') | $1.5534 \times 10^{-20}$ (a') | $1.5534 \times 10^{-20}$ (b) |
| Mean quantum number ($j_r$) | 10.513 | 10.513 | 7.052 | 7.052 |
| Mean quantum ($h\nu$, J) | $3.9547 \times 10^{-21}$ | $3.9547 \times 10^{-21}$ | $2.20268 \times 10^{-21}$ | $2.20268 \times 10^{-21}$ |
| Energy density ($g_t/a^3$, J/m$^3$) | $1.97236 \times 10^7$ | $7.28400 \times 10^6$ | $0.36828 \times 10^6$ | $1.00107 \times 10^6$ |
| Frequency ($\nu$, Hz) | $5.96831 \times 10^{12}$ | $5.96831 \times 10^{12}$ | $3.32421 \times 10^{12}$ | $3.32421 \times 10^{12}$ |
| Wavelength (m) | $5.02308 \times 10^{-5}$ | $5.02308 \times 10^{-5}$ | $9.01846^{-5}$ | $9.01846 \times 10^{-5}$ |
| $\lambda/2\pi r$ | $1.24702 \times 10^4$ | $8.93525 \times 10^3$ | $8.24669 \times 10^3$ | $1.15092 \times 10^4$ |
| $n_t^3 \times j_r^2$ | $1.0616259 \times 10^8$ | $2.885798 \times 10^8$ | $2.885798 \times 10^8$ | $1.0616259 \times 10^7$ |

See Table S1 (Carnot8/Cal program) in Supplementary Materials for a program to calculate all values in Table 1.

- The ratio for wavelength of virtual quanta and the material radial motion ($\lambda/2\pi r$; $r = a/2$) of approximately $10^5$ for the gases is indicative of the ratio between the speed of light ($\lambda v = r\omega$) and speed of the Brownian spiral of gas molecules. This can be visualized as the frequency of the conjugate quanta being of a similar order to that of the orbital frequency of the molecules but with the photon's impulse cycling on a much longer radius, proportional to the ratio of speeds ($c/v$).

- Table 1 also illustrates the correspondence for both argon (mass 40) and nitrogen (mass 28) of the ratio of the cumulative quantum impulse ($nh/\lambda$) and the dynamic impulse ($\Sigma mv = \Sigma mr\omega$) per molecule. This is a factor near $1 \times 10^{-5}$, the inverse of the ratio of the speed of light to that of the molecules. In calculating translational action of molecules [2] it is necessary to make two corrections. One, a factor of $1/1.09$ corrects root mean square velocity from the Maxwell distribution ($3kT = mv^2$) to mean velocity. The second corrects action for symmetry to avoid double counting of molecules ($1/2$). For cubic translation, this is an overall factor of $1/10.2297$. This correction then allows the entropy calculated to match that for third law experiments in the literature. This correction factor ($z_t$) was initially established empirically [1], then interpreted rationally [2]. Overall, it allows the density of quanta needed to sustain the system to fall by a factor of $2.3205$.

- Another possible source for lack of correspondence in matching action impulses between quanta and molecules is that phase space for position and momentum can never exactly match true action space. The ideal coordinate system may not be Cartesian phase space since this separates variables ($mv$ and $r$) that must be combined when quantized. A radial or polar system ($r$, $\phi$, $\theta$) is needed [7], but one that recognizes that changes in position in 3-D is absolutely quantized as jumps in the space of objects from one locus to another. There is no such thing as a smooth curve in nature for translation of rotation except by perception within the space of views, as explained a century ago by Jean Nicod [10].

This analysis gives good correspondence between potential impulse rates of quantum and dynamic molecules in the field. For the nitrogen molecule there is clearly an effect of rotation, responsible for approximately 20% of the quantum energy density. The number of quantum states for translation increases with altitude at lower temperatures and pressure, but that for rotation does the opposite.

It is unfortunate that the text of Carnot's monograph [4] was lost for many years after his premature death from cholera in 1832. Caloric was clearly regarded by Carnot as causal for the power (puissance motrice) of heat engines. Lord Kelvin and Clausius assumed for convenience that the heat consumed in Carnot's cycle reappeared directly as the external work performed by the heat engine. This was despite Clausius originally speculating that heat was concerned with performing internal work of the fluid needed to support external work. Later, this process of internal work that he named the system's ergal, suggesting this term could replace Rankine's new definition of potential energy [2], was neglected; the heat consumed was judged to be transformed directly to external work, not that it was required internally to support the external work, as found in the Carnot cycle [2].

The term radial action indicates the complementary relationship between maintaining the action of a particle and the action of its complementary quantum of similar frequency. The wavelength of the quantum is always much larger than the radial motion of the particle. However, it is relevant that impulses from particles such as quanta and molecules exert torques counted as kinetic energy proportional to their radius of action. The impulse of the quanta on their conjugate wavelength exerting torque is equal to that of the molecules on their radius ($r$).

*3.2. Thermodynamic Stabilization with Altitude for Atmospheric Gases by Quantum Fields of Molecules in Air*

Assuming that translational action of molecules in air refers to non-linear or curved motion in the dimensional form $mr^2\omega$ as used for calculating translational entropy in

heat engines, shown in Table 1, the following relations by equating thermodynamic and gravitational pressure with altitude can be derived. In Equation (8), $a$ represents the mean length of the side of a cube occupied by each different molecule, and $r$ represents the mean radial separation which is half of that value. Then $M$ is equal to N$m$, the total mass of $n$ molecules in the atmospheric column above a square with base of side $a$ cm, of weight N$mg$, assuming the value of gravity ($g$) is invariant in the troposphere. It is assumed that the inertial force $mr\omega^2$ provides the internal pressure on the six faces of the cube of side $a$, tending to equilibrate with the gravitational pressure or weight ($Mg$) per unit area ($a^2$) of the atmosphere.

$$p = \frac{kT}{a^3} = \frac{mr^2\omega^2}{3a^3} = \frac{Mg}{a^2} = mr\omega^2/6a^2 \qquad (8)$$

Thus, the instantaneous gravitational force on average necessarily exerted by each molecule is one-sixth of the centrifugal or inertial force $mr\omega^2$ exerted in each six-faced cell of side $a$ by the translational motion of each molecule. The primary gravitational pressure from the weight of air is exerted once only downwards and not to all six cardinal points. The thermodynamic relationship is statistical according to the Maxwell–Boltzmann distribution, with the molecular velocities having values statistically varying around the root mean square velocity which is characteristic of the temperature. Incidentally, the hydrostatic or isobaric requirement that the pressure is a function of density is only true for an isothermal atmosphere. In a real atmosphere with a tropospheric temperature gradient with an altitude of a little more than 6.5 K per km [7], pressure also varies as a function of temperature.

The common opinion that the decline in temperature with altitude is an adiabatic response to expansion of atmospheric gases, as may occur in a cylinder undertaking external work with increasing volume, was challenged [7]. Instead, the decline in temperature with altitude indicates the operation of the virial theorem for gases of differing gravitational potential and thermal energy. Changes in potential energy indicate changes in both quantum state and field as well as changes in kinetic energy. According to the virial theorem, the absorption of a quantum of gravitational energy causes a decrease in kinetic energy of the same magnitude, meaning that the increase in potential energy is twice either the decrease in the kinetic energy or the increase in the field gravitational energy. Thermal and gravitational fields are quite separate but tend to equilibrate.

A simple formula to obtain the theoretical temperature gradient [7] for a quasi-equilibrium distribution is $mgh_n$ is equal to $nk\delta T$, yielding a lapse rate of $\delta T/h_n$ or $mg/nk$ rather than $mg/C_p$, where the divisor is the heat capacity of air at constant pressure. This new formula gives a lapse rate of temperature with an altitude of 6.9 C per km, close to the observed value and is only slightly reduced by heat released on condensation of water vapor. In the formula, $m$ is the mean molecular weight of the gases in Daltons (28.97 for air) number-density weighted, where n indicates the degrees of freedom of action or kinetic motion able to contain heat, usually each $k/2$, although quantum effects of vibration can modify this freedom affecting n. For monatomic molecules such as argon, n is three. For diatomic molecules such as nitrogen and oxygen in their ground states it is five, and for simple polyatomic molecules found in the atmosphere it is six at ambient temperatures. However, the actual temperature gradient with altitude will be a cumulative variable determined by the complex properties of other gases in the atmosphere, their mixing ratios, and other local environmental factors such as temperature that may affect the vibrational heat capacity or quantum state. This is not a significant issue for the major diatomic gases in the atmosphere of Earth but would be on Venus [7] with its surface temperature in the vicinity of 740 K, with carbon dioxide as the major gas.

Action mechanics [1,2] combines the macroscopic variations in volume (N$a^3 = 8$N$r^3$) or density and temperature of molecules as action ($mrv\delta\phi$), illustrated in Section 3.1. In Table 2, estimates are given for thermodynamic properties of atmospheric $N_2$ as translational ($n_t$) and rotational ($j_r$) molecular action. Activation of vibrational states for $N_2$ in the atmosphere are negligible, as shown previously [1], given the high frequency and energies involved.

If required, vibrational Gibbs energy can be estimated from the statistical component of vibrational entropy together with the zero-point energy $Nh\nu/2$ of 14.115 kJ per mol. More than 75% of the energy content indicated for $N_2$ is required to sustain its translational Gibbs field, shown in Table 2, with resonant quanta in the range 3.8 to $2.2 \times 10^{-22}$ J of frequency $5.7 \times 10^{11}$ to $3.32 \times 10^{11}$ Hz and wavelengths from 523 to 904 μm. $N_2$ contributes almost nothing [1] to the thermal emission to space in the infrared and far infrared up to 100 μm wavelengths, unlike water and other greenhouse gases.

**Table 2.** Tropospheric variation with altitude in molar entropy ($S$) and Gibbs function ($G/T$) for the major tropospheric gas $N_2$ (78.04%) in the Model 6 US standard reference atmosphere.

| Alt km | Temperature K | Pressure Atm | $S_t$ J/mol/K | Mean Level $n_t$ | $ST$ Trans. kJ/mol | Mean $h\nu(\times 10^{-22}$ J) Quanta | $S_r$ J/mol/K | Mean Level $n_r$ | Mean $(\times 10^{-21}$ J) Quanta $h\nu$ | Total $(S_t + S_r)T$ kJ/mol | Gibbs $G$ kJ/mol |
|---|---|---|---|---|---|---|---|---|---|---|---|
| 0 | 288.2 | 1.000 | 151.76 | 190.69 | 43.74 | 3.809 | 40.92 | 10.05 | 1.94900 | 55.556 | −61.225 |
| 1 | 281.7 | 0.886 | 152.28 | 194.71 | 42.90 | 3.659 | 40.73 | 9.94 | 1.91743 | 54.370 | −60.284 |
| 2 | 275.2 | 0.785 | 152.81 | 198.93 | 42.05 | 3.511 | 40.54 | 9.82 | 1.88723 | 53.210 | −59.316 |
| 3 | 268.7 | 0.692 | 153.36 | 203.34 | 41.20 | 3.446 | 40.34 | 9.70 | 1.85625 | 52.047 | −58.340 |
| 4 | 262.2 | 0.609 | 153.91 | 207.92 | 40.36 | 3.223 | 40.14 | 9.59 | 1.82303 | 50.880 | −57.365 |
| 5 | 255.7 | 0.534 | 154.48 | 212.73 | 39.50 | 3.083 | 39.93 | 9.47 | 1.79095 | 49.771 | −56.385 |
| 6 | 249.2 | 0.466 | 155.08 | 217.89 | 38.65 | 2.946 | 39.71 | 9.35 | 1.75808 | 48.544 | −55.405 |
| 7 | 242.7 | 0.401 | 155.78 | 224.09 | 37.81 | 2.802 | 39.49 | 9.22 | 1.72675 | 47.394 | −54.444 |
| 8 | 236.2 | 0.352 | 156.36 | 228.80 | 36.93 | 2.680 | 39.27 | 9.10 | 1.69318 | 46.194 | −53.475 |
| 9 | 229.7 | 0.302 | 157.00 | 235.26 | 36.06 | 2.545 | 39.04 | 8.97 | 1.66066 | 45.028 | −52.458 |
| 10 | 223.3 | 0.262 | 157.59 | 240.92 | 35.19 | 2.426 | 38.80 | 8.85 | 1.62623 | 43.854 | −51.471 |
| 11 | 216.8 | 0.224 | 158.28 | 247.67 | 34.59 | 2.319 | 38.56 | 8.72 | 1.59251 | 42.672 | −50.479 |
| 12 | 216.7 | 0.192 | 159.55 | 260.63 | 34.57 | 2.203 | 38.55 | 8.71 | 1.58327 | 42.928 | −50.737 |

Data were calculated [1] from molecular properties, temperature, and pressure values shown using Equation (7) relating to translation and rotation only. A capable program to compute all relevant quantities in cgs units (1 Joule = $10^7$ ergs) is available as Table S2 Entropy8/Cal in Supplementary Materials. $S_t$ and $S_r$ are entropies per mole for translation and rotation, respectively.

However, both the translational and rotational fields contain large numbers of quanta, with wavelengths longer than 520 μm and 100 μm, respectively. It was described in detail how these values are easily calculated and employed in refs. [1,2,6]. In our opinion, too much emphasis is placed on the calculus of the Maxwell relations in teaching thermodynamics and too little on mechanical methods of computation for real-world molecules. Equation (7) has the advantage of yielding mean quantum values, with the radial action needed to provide torques sustaining the dynamic motion of the molecules indicated by the temperature ($T$). The use of the form $ST = H − G$, where $H$ is enthalpy and $G$ is Gibbs energy, emphasizes the fact that entropy or its product with temperature is not a single property that stands alone but consists of the sum of the internal energy or enthalpy and a statistical field containing quantum information required for the molecular configuration of the system.

Given that the Gibbs energy, defined by Equation (7), is regarded as responsible for the configuration of the molecular system as well as positions of equilibrium in chemical reactions [6], it is surprising that this field receives so little attention. In proposing its importance in the atmosphere, what is the likelihood that this energy field for translation and rotation can also contribute to radiation lost to space? Much information is collected on surface radiation from Earth by satellites, but this is usually restricted to the thermal region of vibrational infrared wavelengths of less than 100 μm than of longer microwaves or radio waves that will be released from the upper atmosphere when less caged by matter. Any such field energy losses will be continuously replenished with thermal energy from

the surface, as predicted by the equipartition principle, subject to quantum restrictions at lower temperatures.

For translation and rotation of $N_2$ molecules, Table 2 shows estimates with altitude for entropy per mole ($S$), absolute Gibbs quantum state levels ($n_t$), the total heat energy required to reach the state ($ST$), and the mean value of virtual field quanta ($hv$). Vibrational energy for $N_2$ is negligible. To estimate peak wavelengths of quanta in the field energy, the virtual quantum value ($hv$) is divided by $h$ to obtain frequency ($v$) then inverted and multiplied by the speed of light $c$ to obtain the wavelength. Thus, at the surface temperature and pressure, it can be shown that $1.14833 \times 10^{26}$ quanta per mol of peak translational energy $3.809 \times 10^{-22}$ J support the kinetic activity of $N_2$ with a frequency of $5.74848 \times 10^{11}$ (574.849 GHz) with a peak frequency at a wavelength of 521.52 μm. At 12 km of altitude, the peak wavelength is almost twice as long at 901.704 μm, though there are more quanta per molecule. The table also shows absolute values for the Gibbs field energy per mol, a property that is always negative and becomes more so as spontaneous processes occur that increase the entropy.

Gibbs energy is often referred to as Gibbs free energy or chemical potential. It can be made positive in value when expressed in the usual form at atmospheric pressure as $G = H - ST$, but it must be appreciated that $ST$ contains Gibbs energy ($G$) and enthalpy ($H$) or internal energy ($E$). Little attention is paid to this major reservoir of internal quantum state energy in the atmosphere, with the kinetic energy and macroscopic pressure–volume work at atmospheric pressure usually given prominence.

*3.3. Vortical Action as High-Level Atmospheric Thermodynamics in Anticyclones*

Our hypothetical introduction of vortical energy [2] for a molecular field in concerted motion is estimated by a similar method to that used for the Carnot cycle (Figure 1). The translational action ($@_t$) of air molecules in anticyclones concerted in motion as wind velocity is estimated as $mrv$ from knowledge of the mass of a material particle, the radial separation $r$ equal to $d/2$, where $d$ is the diameter of the anticyclone (Figures 1 and 2). The number of quantum levels is estimated using division by the reduced Planck's constant of action ($h/2\pi = 1.054 \times 10^{-34}$) ($n_t = mrv/\hbar$), with a symmetry factor of two for symmetrical partners. The Gibbs vortical energy per matter cell $a^3$ is then estimated using the logarithm of the quantum number multiplied by the appropriate torque factor. Frequency and wavelength are then easily determined. As with the Carnot cycle, the ratio of mean quantum wavelength and that of the radius to the center of the anticyclone is of the same magnitude as the ratio of the mean velocity of the molecule to the speed of light. The total vortical Gibbs energy is obtained from the product of number of molecules per cubic meter and the number of molecules.

Figure 1 integrates the vortical rotational energy in anticyclones and cyclones with the global Trenberth heat flow budget [11] for radiation. The Kiehl–Trenberth budget proposes that 332 W per $m^2$ of downwelling radiation is returned from the atmosphere, which explains the blackbody temperature of the Earth's surface. It is proposed that instead of net radiation from higher in a colder atmosphere to the surface, vortical action in anticyclones is generated as work processes in air facilitated by Coriolis accelerations in each hemisphere. This work process requires significant absorption of heat radiated from the surface in greenhouse processes involving mainly water and carbon dioxide without rises in temperature. As shown in the Carnot cycle, any increase in freedom of relative translational motion of molecules increases the heat capacity of the gas phase. For anticyclones, this allows turbulent friction processes nearer the surface to release heat in the boundary layer of the lower atmosphere ($\Sigma hv$) to the extent (h = a + e + c + g = $\Sigma hv$) of approximately 330 watts per $m^2$ [2] as a global average rather than by radiation from a colder atmosphere to the surface, in accordance with the second law of thermodynamics. The decreasing wind speed near the surface regarded as vorticity represents the loss of power with wind speed, warming air, and causing spectral radiation proportional to temperature.

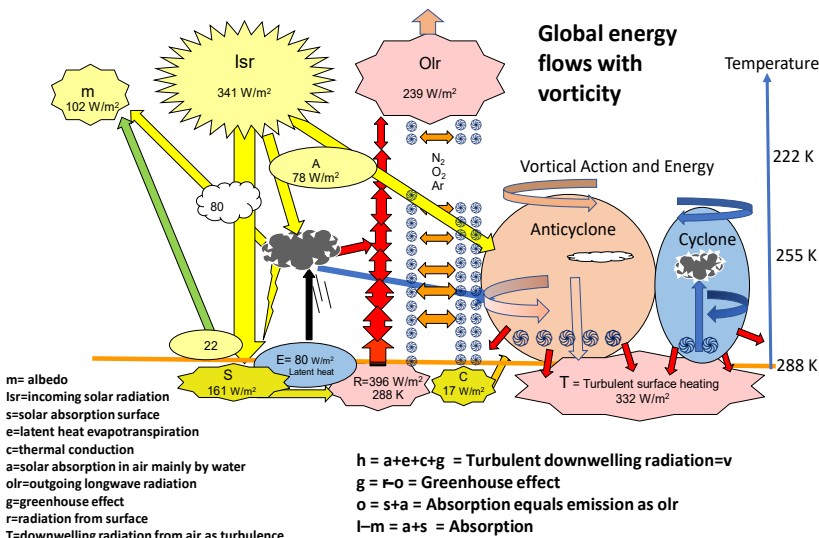

**Figure 1.** Global model for the role of vortical action in anticyclones and cyclones, coupled with the Kiehl–Trenberth heat budget [11]. Vortical energy quanta ($\Sigma h\nu$) for warming the Earth's surface are calculated as shown, supporting the vortical action of molecules per m$^3$ circulating around the higher-pressure center, and acting as rotational winds of low curvature in the plane of the Earth's surface. Note that all symbols (m, s, e, c, a, g, T, h, o, i) or their upper-case versions are specific to this figure.

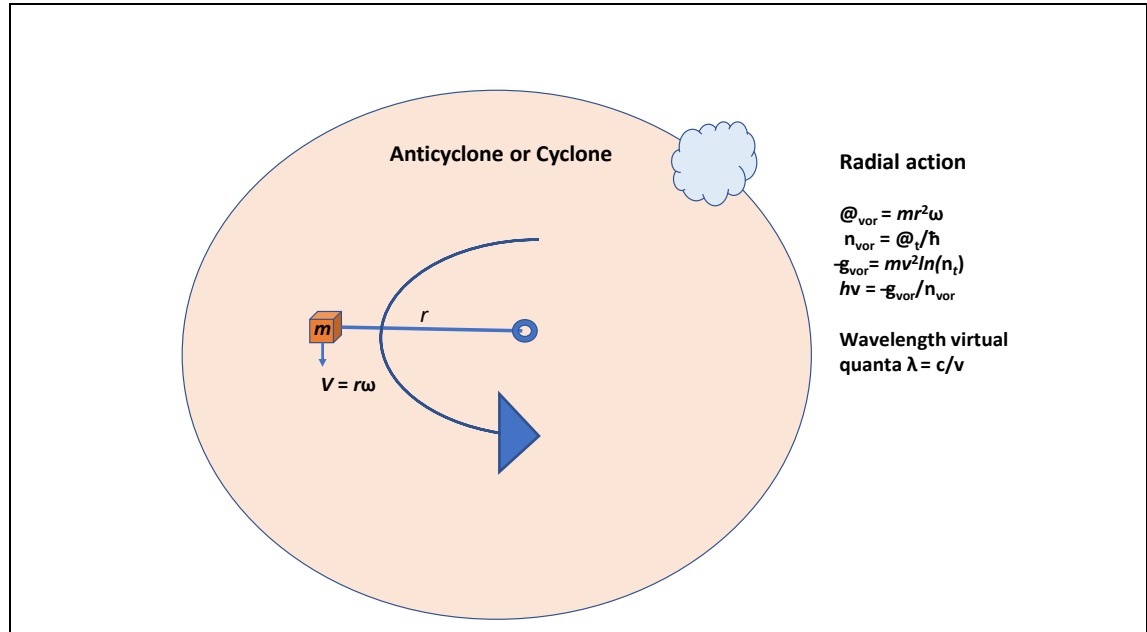

**Figure 2.** Vortical action ($@_w$) and entropy in anticyclones. Action for a cubic meter of wind molecules is computed to generate vortical Gibbs energy at *r* cm from the origin. The associated quantum field has a wavelength $\lambda$ so that c/*V* is of the same order as $\lambda$/*r*.

Our model for greenhouse warming in Figure 1 (g = r − o) involves the difference between blackbody radiation (r = $aT^4$) from the surface and outgoing longwave radiation (o), with the atmosphere warmed by solar absorption by water in air (a), latent heat of evapotranspiration at the surface followed by condensation under convection (e), thermal conduction from the surface (c), and the greenhouse effect itself (r − o). The work of vortical action and energy (v) provides a mechanism for turbulent release of heat as radiation near the surface. No conflict with the second law of thermodynamics is required, solving the

objection to net downwelling radiation from a colder source. As shown in Table 3, a wind speed of 10 m per sec contains a vortical energy of $1.47 \times 10^3$ J per $m^3$ of air in wind, many times greater than the kinetic energy, with an additional 2.4 MJ per $m^3$ of thermal energy required for air to be heated from 0 K to 298 K [1].

**Table 3.** Kinetic and vortical energy impacting a wind turbine with 1.5 MW power output.

| Wind Speed $V$ (m/s) | Kinetic Energy/s 83 m Diam J | Kinetic Energy /Blade-Area/s J | Vortical Pressure, J/m$^3$ | Vortical Power for Blade Area Watts | Power Estimated by Radial Action Model Watts |
|---|---|---|---|---|---|
| | | | | | At $\lambda$ = 9, pitch $\theta$ = 55° |
| 5.0 | $0.21670 \times 10^6$ | $8.4066 \times 10^3$ | $0.36107 \times 10^3$ | $0.33038 \times 10^6$ | $0.031168 \times 10^6$ |
| 10.0 | $1.7336 \times 10^6$ | $6.7253 \times 10^4$ | $1.47258 \times 10^3$ | $2.69482 \times 10^6$ | $0.40541 \times 10^6$ |
| 15.0 | $5.8509 \times 10^6$ | $2.2698 \times 10^5$ | $3.35055 \times 10^3$ | $9.19727 \times 10^6$ | $1.54381 \times 10^6$ |
| 20.0 | $1.3869 \times 10^7$ | $5.3802 \times 10^5$ | $6.00353 \times 10^3$ | $21.9729 \times 10^6$ | $3.86798 \times 10^6$ |

Rate of kinetic energy calculated in Table S3 (Turbine5/Cal program) in Supplementary Materials; $\lambda$ is tip-speed ratio = $R\Omega/V$.

The theory of vortical action and energy predicts that the troposphere at the Earth's surface has an additional reversible thermodynamic heat capacity of significance for meteorology. The cyclic nature of anticyclones or cyclones and their transit across the Earth's surface cause rapid changes in weather, a topic of daily interest and conversation everywhere. Seasonal effects such as monsoons have even more striking effects, as those who have experienced the monsoon breaks of northern Australia know well. Vortical action, like all forms of molecular action, must be associated with an extra reversible heat capacity as field energy quanta. One area of changing human endeavors possibly developing without clear knowledge of consequences is that of utilizing wind power as a renewable source of energy and electricity. Will this change be without serious environmental consequences, as governments everywhere now assume? Managing such risks is an important area for regulation that must be based on the best scientific knowledge.

*3.4. Estimation of Power Produced by Wind Turbines from Vortical Energy in Anticyclones*

In the study of the thermodynamics of heat engines, theory has usually followed practice. Heat engines powered with steam invented by practical trial and error by Newcomen, Watt, and others started the industrial revolution well before the theoretical thermodynamics of reversible heat engines was initiated by Carnot, Joule, Kelvin, and Clausius. Windmills are an even more ancient form of heat engines, though are dependent on the vagaries of weather with limited reliability. The scale of application of modern wind turbines in the past 20 years has amplified the possible environmental significance of these machines, just as much as other heat engines operating with similar mechanical principles. The focus in this article is on the driving force of quantum fields of energy as the cause of kinetic motion in engines, observable as torques or forces with leverage of differing radial advantages. This justifies analysis of wind and wind turbines as operating with this vortical energy, a factor so far neglected and possibly overlooked in solving the Navier–Stokes equations. The Gibbs field of associated driving wind power is the topic of analysis in this section.

In our novel theory estimating wind power [8] based on the rate of impulsive action, torque is estimated for both windward and leeward surfaces of wind turbines. Using the difference in these torques, power can then be estimated using the angular velocity of the turbine rotors ($\Omega$). Current blade element momentum theory estimates the potential wind power available by using the rate of kinetic energy of wind flowing through the area swept by the rotor blades, despite most of the air passing through not impacting the turbine blades. Figures 1 and 2 show winds in anticyclones, and the impulsive power of

the energy in air in vortical action striking turbine blades is regarded a better model of turbine function, shown in Figure 3.

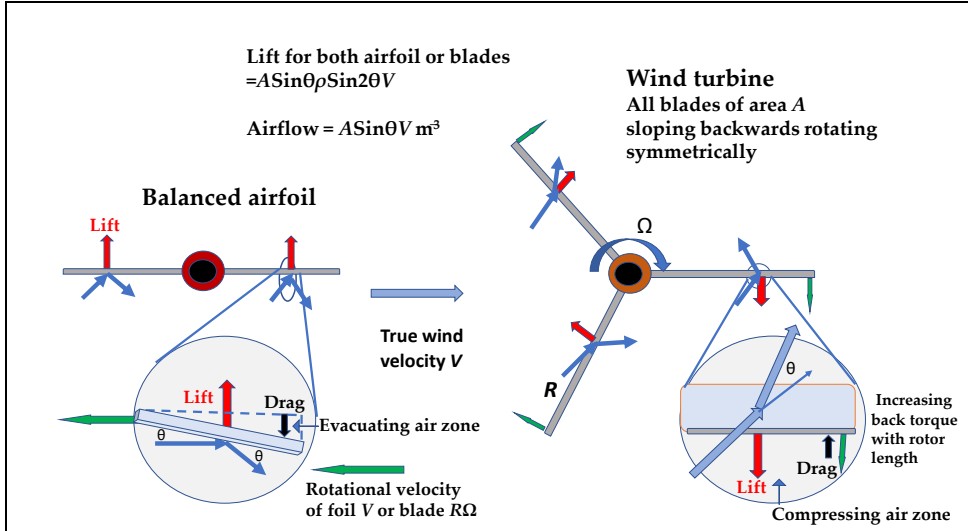

**Figure 3.** Contrasts in function of balanced air foil pairs and wind turbine blade aerodynamics. Air behind the foils moving at forward speed $V$ is depleted, equivalent to the volume of air deflected under the wings ($A\text{Sin}\theta V$/sec, $A$ area and $\theta$ angle of wind deflection) and replenished immediately by the drag from air dropping from above. Air in front of the rotor blades is compressed by rotation, exerting a back torque and decreasing wind power, particularly at a longer radius ($R$). Air foils are normally horizontally balanced in flight, preventing rotation or rolling.

Radial action estimates at different speeds for a wind turbine with an 83 m rotor diameter are shown in Table 3. The estimate for power as rate of kinetic energy may be considered as giving power according to the Betz limit of 0.59. Unsurprisingly, given the lack of adjustments for tip speed ratio ($\lambda$) and blade angle, there is disagreement with that estimated by radial action, particularly at low wind speed. This value is also corrected to the kinetic energy actually impacting the blades in column three, reducing the power estimate to a fraction of that actually achieved by radial action. The idea that the wind power is derived from the quantity of kinetic energy of the wind, except to provide the molecular pressure sustained in turn by the field quanta, is rejected. Table 3 compares the power available from kinetic energy and from vortical potential energy for a parcel of air per cubic meter located 1000 km ($r = 10^6$ m) from an anticyclone center (Figures 1 and 2). The analysis considers the wind impacting the rotor blades as well as that passing through the circle of blade rotations. Only in the latter case does the kinetic energy available to the blades exceed the radial action estimate. By comparison, the maximum vortical power estimated is more than five times greater and is restricted to the blade area.

In our new Newtonian action method [8] to estimate wind power, lift on turbines is shown as a function of the rate of reflected momentum of air volume, proportional to sin2$\theta$, varying with twice the angle of wind incidence ($\theta$) on rotor blades (Figure 3). Although blade element momentum air foil theory is usually applied to wind turbines, this may be inappropriate given that the blades compress air while transferring momentum in the direction of rotation; by contrast, the air foil in flight creates an evacuated zone above the foil at a rate equivalent to the air volume deflected underneath. Normally, air foils do not rotate into the air. In the radial action model, this vacuum is filled continuously from the massive 10 ton per square meter column of air above. Similar equations used for estimating power in wind turbines can be applied to air foils, estimating lift for aircraft and the drag force by calculating the momentum of air falling into the space above the foil.

More recently, in our research paper still under review advancing the article lodged on arXiv [8], the relationship $A\text{Sin}\theta\rho\text{Sin}2\theta V$ in Figure 3 for lift was found applicable to

both air foils and rotor blades. Here, the turbine blade or air foil area (*A*), angle of air flow incidence (θ), air density (ρ), and air stream or wind velocity (*V*) are the key factors. Though surprising, the Newtonian radial action model equations seem to be valid for both wind turbines and air foils lifting aircraft.

Shown in Table 4, the virtual Gibbs field contains approximately 70 times as much energy as the density of kinetic energy of the molecules. This field configurational energy sustains the kinetic energy. The idea that all air passing through the circle swept by the turbines never contacting the turbine blades could have sufficient solidity to have a significant effect on power output challenges reason. Instead, impulsive radial action must generate torques strictly exerted on the blades. Some proposals regarding the nature of vortical energy per paired molecule are given in Table 4. Using the torque generated per molecule ($kT = mv^2$) at a 1000 km radius, estimates for vortical energy available are almost two orders of magnitude greater than that of kinetic energy.

**Table 4.** Vortical energy properties for a 1.5 MW wind turbine.

| Wind Speed (m s$^{-1}$) | Vortical Action (*mrv*/2 =@$_v$)/ Molecule J.s, $\times 10^{19}$ | Quantum Number n$_{vor}$ $\times 10^{-15}$ | 1-D Torque mv$^2$/Mole- Cule $\times 10^{-24}$ J | Vortical Energy /Molecule [($mv^2$)ln(n$_{vor}$), $\times 10^{-23}$ J | Vortical Energy J/m$^3$ | Vortical Wave- length $\times 10^{-12}$ m | Kinetic Energy J/m$^3$ | Ratio Vortical/ Kinetic Energy |
|---|---|---|---|---|---|---|---|---|
| 5.0 | 1.2108 | 1.14812 | 1.2108 | 4.2812 | 1062.024 | 5.3272 | 15.313 | 69.355 |
| 10.0 | 2.4215 | 2.29615 | 4.8430 | 17.1297 | 4332.826 | 2.6627 | 61.250 | 70.737 |
| 15.0 | 3.6823 | 3.49168 | 10.8968 | 38.998 | 9864.433 | 1.7786 | 137.813 | 71.758 |
| 20.0 | 4.8430 | 4.09488 | 19.372 | 69.862 | 17670.951 | 1.1643 | 245.000 | 72.127 |

Radius = 1000 km; SI units (kg-m-s); symmetry factor for action is two.

A feature of the classical action model for quantum fields is that it allows absolute values of Gibbs energy content per molecule to be calculated rather than transitional values (δ*g*).

For example, for wind at 10 m per sec, the mean virtual quantum (*h*v) has the value of $7.46018 \times 10^{-38}$ J with a wavelength of $2.6627 \times 10^{12}$ m. This wavelength is more than $10^6$ greater than the corresponding material radius. This means that the curvature of the oscillating longitudinal motion is relatively linear and the action velocity for the molecules is approximately $10^6$ less than the velocity of light. Differences in Gibbs energy per molecule with wind speed are easily calculated. Methods are suggested in our articles [2,6,8] to test the vortical energy field hypothesis using appropriate sensors, allowing release of this energy under turbulent conditions. If confirmed, this could be an important source of regional warming and land dehydration [8], possibly raising fire risk from wind farms. A 100 MW windfarm could raise the temperature of air downwind by the turbulent release of vortical energy up to 2 °C, increasing evapotranspiration for many kms. This prediction [8] is recommended to be tested as a matter of diligence regarding the location of windfarms.

*3.5. Power in Tropical Cyclones Estimated by Heat from Volatilization on the Ocean Surface and Convective Condensation of Water at the Eyewalls*

Tropical cyclones are known to be fully energized by the heat of vaporization of water at warm ocean surfaces in low latitudes. Condensation by strong spiral convection near the eyewall releases radiant infrared energy as directly resonant quanta from H-bonding between water molecules in the formation of water droplets [12]. Yet, it is also well known that the kinetic energy of the cyclonic motion is only a small percentage of the total energy and power dissipated in the path of a cyclone's trajectory.

Vortical action generated at the cyclone's eyewall nearer its center provides the direct connection needed between heat of condensation and the power of the cyclone, with the kinetic pressure and torque being sustained by the impulses of the Gibbs field energy. The

field energy pressure is a large multiple of the kinetic energy pressure and can sustain the torques required, at least while the cyclone continues to evaporate water from the sea surface. On land, the energy of the cyclone, no longer sustained by evaporation from the ocean, is soon dissipated by frictional turbulence releasing heat to the surface atmosphere. Such intense regional warming on land is an anecdotal feature after tropical cyclones on land.

It is assumed that specific radial action ($r^2\omega$) will be approximately constant across the cyclone, given the inverse square radial distribution of radiation from convective condensation at the eyewall, generating a radially acting Gibbs energy field as $mv^2\ln[n_v]$ as shown in Table 5. Furthermore, considering the cyclone as simultaneously rotational and convective, conservation of angular momentum will ensure constant $\Sigma mr^2\omega$ ensuring intensified velocity ($r\omega$) nearer the eyewall. The heat of vaporization at an estimated rate matches the vortical Gibbs energy and power generated, as shown in Table 5. Vortical entropic energy is directly provided by infrared radiation from condensing water in the convective eyewall of a tropical cyclone. The model shown in Figure 4 exhibits vortical action and energy, as shown in Table 5. Note the 70-fold ratio of energy density in the Gibbs field compared to that of kinetic pressure, similar to that observed for wind driving turbines to develop electrical power (Table 4).

**Table 5.** Vortical entropic energy from infrared radiation of water condensation at the convective eyewall of a tropical cyclone.

| Radius km | R | 50 | 100 | 150 | 200 | 250 | 300 | 350 | 400 | 450 | 500 |
|---|---|---|---|---|---|---|---|---|---|---|---|
| Speed m/sec | $r\omega$ | 100 | 50 | 33.33 | 25 | 20 | 16.67 | 14.29 | 12.5 | 11.11 | 10 |
| @$_v$ J-sec × $10^{15}$ | $n_v = mr^2\omega/2\hbar$ | 1.481 | 1.481 | 1.481 | 1.481 | 1.481 | 1.481 | 1.481 | 1.481 | 1.481 | 1.481 |
| Torque/molecule Joules × $10^{-24}$ | $mv^2 = mr^2\omega^2$ | 484.3 | 121.1 | 53.81 | 30.27 | 19.372 | 13.453 | 9.884 | 7.567 | 5.979 | 4.843 |
| −Gibbs/molecule Joules × $10^{-22}$ | $mv^2\ln[n_v]$ | 167.940 | 41.985 | 18.660 | 10.496 | 6.717 | 4.665 | 3.427 | 2.624 | 2.073 | 1.679 |
| Kinetic/molecule J/m$^3$ × $10^{-3}$ | $p = 1.5kT/a^3$ | 6.125 | 1.531 | 0.681 | 0.383 | 0.245 | 0.170 | 0.125 | 0.096 | 0.076 | 0.061 |
| −Gibbs J/m$^3$ × $10^5$ | $p = J/m^3$ | 4.247 | 1.062 | 0.472 | 0.265 | 0.170 | 0.118 | 0.087 | 0.065 | 0.052 | 0.046 |
| Ratio pressures | Gibbs/kinetic | 69.3 | 69.4 | 69.3 | 69.2 | 69.4 | 69.4 | 69.6 | 68.7 | 68.4 | 75.4 |

1 atm = $1.013 \times 10^5$ J/m$^3$; the integrated thermal input for 25 mm evaporation a day sustains a 500 km radius cyclone 10 km high containing $8.794743 \times 10^{19}$ J of energy, approximately twice the daily input; see Table S4 (Tropcyc2/Cal program) in Supplementary Materials for program.

As discussed in Section 3.2, the pressure equation underestimates the kinetic energy density by 50%, adjusted in Table 5. The greater the vortical energy in the field, the more negative the Gibbs energy of the molecules in the field, as the potential to take up thermal energy in the system is exhausted.

This vortical version of the dynamics of a tropical cyclone may be more consistent with the rotating convective model now accepted for best explaining destructive cyclonic intensification [13]. Previous opinion [14,15] favored intensification as caused by large latent heat fluxes from the surface in the core region and that an intensive evaporation-wind feedback process (WISHE) was not needed. Vortical energy would be stored cumulatively during convective processes causing condensation nearer the eyewall or walls if serried in convective configuration. The vortical inertia of the cyclone would require a continuous feed of thermal energy from evaporation and condensation of water, but this can be modeled as intensifying as updraft increases, shrinking the radius and providing a higher pressure of vortical field energy nearer the eyewall with higher wind speeds. The model data in Table 5 represent a snapshot of the process at one stage in time. Researchers in the dynamics of tropical cyclones are invited to include vortical energy in their models.

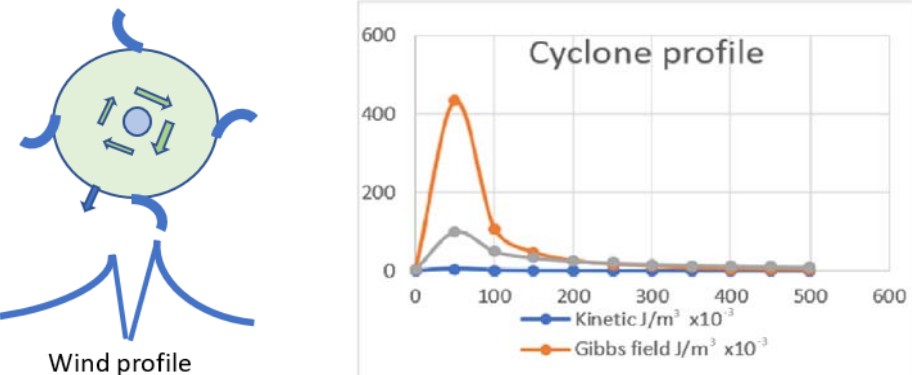

**Figure 4.** Cyclone radius shown in km (*x*-axis) of a tropical cyclone at 17° S latitude, showing greatest intensity to the southwest and accentuated by the Coriolis effect deflecting the cyclone to the southwest. Wind speed (*y*-axis m/s, see Table 5) shown in grey peaks at the eyewall; the kinetic energy pressure or torque pressure in blue appears minor, but its intensity is maintained by the much larger Gibbs field energy content.

This ability to explain the intensifying power of tropical cyclones quantitatively is a major point in favor of the vortical action and energy theory.

## 4. General Discussion

The scientific strategy in this article is based on results from our previous analyses of the action, entropy, and Gibbs energy of the atmosphere [1] and of the Carnot cycle [2], concluding that quantum fields of thermal energy sustain the internal energy and configuration of molecules. Simple methods [6] to calculate the action, entropy, and Gibbs field energy from molecular properties of gases such as hydrogen, water, and ammonia strengthen the value of this approach, providing new insights for catalytic transitions and equilibrium states, where Gibbs field energies for reactants and products are in balance [6]. Wherever molecules exhibit action spatially with respect to others, such as by vibration, rotation, or translation, a field of impulsive quanta sustains a rate of molecular action as torque or kinetic energy. In this article, this mechanism is extended based on relative rates of action giving vortical torques exerted in molecular flows of atmospheric systems that require resonant Gibbs quantum fields. This innovation allows a significant expansion to the reversible heat and work capacity of the Earth's atmosphere, possibly of major significance in climate science consistent with the law of increasing entropy.

One difference between the Carnot heat engine cycle and the higher-level vortical systems is its absence in the ideal Carnot cycle. The ability of a heat engine to reversibly store heat and perform external work depends entirely on the differences in the Gibbs quantum field at the extremes of temperature. The net variation in energy as kinetic work between the source and sink temperatures in the cycle is zero. Although the vortical field energy of anticyclones and cyclones expressed in the Earth's gravitational field is also shown to provide work potential that can decrease Gibbs energy, this was not predicted until considering climate change in our work. The vortical hypothesis provides solutions to questions regarding (i) the warming of the Earth's surface by frictional dissipation of vortical energy in anticyclones and cyclones rather than by downwelling radiation [11] from higher, colder air in the atmosphere; (ii) how vortical energy in anticyclones can power wind turbines by torques exerted only by the air flow impacting turbine blades [8]; and (iii) how the destructive force of cyclones is directly powered by vortical action linked to energy flow from a thermal cycle of solar evaporation of tropical water, with convective condensation at the eyewall explaining the riddle of the negligible kinetic energy in tropical cyclones.

The Bernoulli interpretation of energy in fluids as a continuum with kinetic pressure-related energy is insufficient to explain these phenomena. The governing equations of fluid

motion as formulated by Bernoulli, Laplace, and others proposed no such reversible heat-work process for vortices, favoring absorption and release of heat in adiabatic processes. For streamlines as in a laminar wind flow, the Bernoulli Equation (9) relates kinetic energy ($\rho v^2/2$), the static pressure energy $P$ ($\Sigma mv^2/3 = pV$), and gravitational potential energy, regarded overall in steady flow as constant.

$$\rho v^2/2 + P + \rho gh = K \tag{9}$$

However, the vortical Gibbs energy calculated for wind (Tables 3 and 4) provides a much larger reservoir of thermal energy as a large multiple of the kinetic energy ($\rho v^2/2$) in Equation (9); it would be better to write the following Equation (10) as prevailing in laminar flows of anticyclones.

$$\rho v^2/2 + P + \rho gh - (\Sigma g_t) = K' \tag{10}$$

This equation might form the fundamental basis for extended solutions to the Navier–Stokes equation, taking into account the potential field energy $-(\Sigma g_t)$ able to be released as heat ($+Q_v$) in vorticity caused by frictional turbulence at surfaces (Figure 1), such as a rough landscape, wind turbines, airfoils, and in the destructive dissipation of tropical cyclones on land. This process amounts to an irreversible loss of work potential as surface heat, although the charging of anticyclones in the atmosphere with radiant heat absorbed by water and other greenhouse gases, such as $CO_2$ and $CH_4$ as illustrated in Figure 1, represents physicochemical work processes influenced by Coriolis effects of the Earth's different latitudes rotating at different speeds.

Furthermore, the release of vortical Gibbs energy as heat when air masses collide, or even in flows of sea water, could cause turbulent heat release as in catastrophic bush fires, or encourage biological production in oceanic gyres. Furthermore, some of the global warming as human population increases could be from increasing frictional resistance of the surface in new vertical cities in addition to heat island effects, or in heat released by turbulence downwind from wind farms [8].

Although the large ratio of vortical energy to kinetic energy in cyclones may seem surprising, this increased means of energy storage in air is small compared to the total energy needed to heat and sustain the translational, rotational, and vibrational freedom of air molecules suspended in the Earth's gravitational field. In 2019 it was shown [1] that air at the Earth's surface requires 2.4 MJ of heat per cubic meter to bring it from absolute zero to ambient surface temperatures, estimated for still air in laboratory conditions. The additional energy of $4.2 \times 10^5$ J (0.42 MJ) per cubic meter predicted in Table 5 at the eye of a tropical cyclone is only 5% greater than the total entropic heat content. The vortical inertial energy shown in Table 3 at a wind speed of 15 m per sec needs only $3.4 \times 10^3$ J of quanta per cubic meter, just over one-thousandth the energy stored in air warmed from absolute zero. From this viewpoint, storing extra energy in air as vortical energy on this scale is unsurprising. If experiments show this prediction is true, this will have important consequences for climate science.

## 5. Conclusions

Action in classical mechanics is based on the principle of least action discussed in Section 2.2. This asserts that natural processes occur optimally with the most efficient distribution of field energy possible. In this sense, the principle is the basis of the second law of thermodynamics. In part a statistical result, this article reveals the significance of the quantum fields that support the kinetic motion of all molecular particles not at absolute zero temperature. These fields complement the much slower motion of molecules, ensuring the same torque or rate of action is experienced as temperature for every species of molecule when at equilibrium. The extension of the freedoms of action from vibration, rotation, and translation to vortical action in the airflows, known as anticyclones and cyclones, is a rational process.

However, the rate of action or torque no longer corresponds to temperature as degrees Kelvin. The quantum fields concerned in cyclonic motion are related to the torques ($mv^2$) experienced in the correlated molecular motions. These vortical quanta or Gibbs fields must be calculated in the same way as internal motions, including translation, but should be related to coherent torques rather than absolute temperature in degrees Kelvin. This field of impulsive energy traveling within vortical systems at the speed of light provides a new reservoir of energy from work processes, just as valid as for molecules heated as gases as in the Carnot cycle. This new understanding of these reservoirs for heat or work is predicted to be important in tropospheric climate science as it involves reversibility.

Most important in a scientific debate is that new thermodynamic proposals must be capable of rational experimental testing as recommended by Karl Popper's book on conjectures and refutation [16]. Our general hypothesis that interactions of Gibbs quantum fields with molecular mechanics may apply for a range of atmospheric systems is consistent with Ockham's razor in that multiplicity in different explanations needs to be avoided. Recent literature reviews [17,18] were examined to find similar action approaches linking quantum fields to molecular mechanics without success. Our approach is therefore the only one that utilizes the integrated property of action measured with Planck's quantum of action as a unit in place of the macroscopic properties of temperature and density for this purpose. In conclusion, the following novel proposals are in need of testing and readers are invited to assist in this process.

- The propositions offered here for atmospheric science are testable, both theoretically and experimentally. Certainly, the detection of the very long wavelengths of translational quanta proposed in anticyclonic winds or tropical cyclones is challenging, and new technology is needed. However, those fields proposed in the Carnot cycle are in the microwave region and for the $N_2$ column in the atmosphere are only an order of magnitude longer. There is an acknowledged dearth of efficient detectors for wavelengths greater than far infrared or microwaves.
- A new method to estimate vortical energy is presented based on the action or quantum state of molecules. This is important as it provides a better understanding of how the Earth's surface is heated as measured by local temperature, how wind power for wind turbines is sustained independent of kinetic energy, and how the destructive power of tropical cyclones is generated from the heat of vaporization of tropical sea water. Far more energy is stored as vortical energy than currently assumed based on laboratory measurements of heat capacity supporting internal energy. This means that heat released from turbulence caused by collisions of air masses or released by wind farms must be considered.
- Practical consequences from the separate analyses conducted in this article include (i) a new means to estimate the lapse rate in the atmosphere that can inform climate models; (ii) a method to model heat absorption in the troposphere by greenhouse gases for recycling (Figure 1) by driving vortical friction at the surface boundary layer; (iii) a new means to predict turbulent heat production downwind of wind turbines, suitable for future field studies; and (iv) a better understanding of the heat-work cycle involved in tropical cyclones, with heat radiated by condensation of water at the eyewall powering the vortical action of the cyclone. All of these findings offer opportunities for new means of gathering information by specific testing technologies.
- The quantum field hypothesis challenges the common opinion that heat is no more than the inertial motion of molecules. Instead, that molecular inertia and motion are sustained by field energy is confirmed, consistent with the density of activated quantum states as in Table 1. The smaller magnitude of translational quanta presents a difficulty for confirmation as there are few spectrometers capable of measuring the intensity of radiation at these long wavelengths. This gap in current technology is likely to be overcome in the future.

Given the flow of solar energy through the Earth's ecosystems (Figure 1), there is an average dissipation of 239 W of solar energy per square meter, a minimum rate of entropy

production of 239/6000–239/288 or 0.790 SI entropy units per second, given the 20-fold change in the temperature of emission of outgoing longwave radiation from 6000 K on the sun's surface to a mean temperature of 288 K on Earth. Since work by winds is possible even for a 288 K surface emission, the temperature of outgoing emissions must be colder, perhaps near 220 K, approximately 70 K colder nearer the top of the troposphere. A whole series of heat-work processes intervene between the initial absorption of solar energy and its emission into space, delaying the expression of maximum entropy. As vortical energy is capable of performing work as wind power, this form of negative Gibbs energy cannot be considered as contributing to entropy as disorder, unless all anticyclones on Earth are considered simultaneously.

Linking the separate fields of matter and resonant quanta provides the inertia and configuration of mechanical systems. Some of the problems regarding structure in complex biological systems, such as processes involving nucleic acids and proteins [19], as well as Monte Carlo molecular dynamics simulations [20] used for continuum mechanics of wind power, could be improved if the requisite Gibbs fields for all degrees of freedom could be determined. Although the strange theory of quanta and matter explained by Richard Feynman [21] in the 1980s dealt with some of the paradoxes introduced into physics by quantum mechanics, we hope that on the practical scale of the phenomena examined in this article, useful new understandings can still be obtained using action mechanics as a version of quantum theory of significance for global sustainability. The possibility of quantum entanglements in the troposphere proposed in this article as long range quanta is rightly a subject of current investigation [22,23].

**Supplementary Materials:** The following supporting information can be downloaded at: https://www.mdpi.com/article/10.3390/applmech4020037/s1; Table S1, Carnot8/Cal program; Table S2, Entropy8/Cal; Table S3, Turbine5/Cal; Table S4, Tropcyc2/Cal; these four microcomputer programs and some data outputs are included.

**Author Contributions:** Conceptualization, I.R.K. and M.H.; methodology, I.R.K.; software, I.R.K.; validation, M.H.; data curation, I.R.K.; writing—original draft preparation, I.R.K.; review and editing, M.H. and I.R.K. All authors have read and agreed to the published version of the manuscript.

**Funding:** This research received no external funding.

**Data Availability Statement:** All data are given in the article or in the Supplementary Materials.

**Acknowledgments:** We are grateful to our host institutions for general support.

**Conflicts of Interest:** The authors declare no conflict of interest.

## Glossary

*Defining the dimensions of action and related energy functions*

The physical property of action, designated here [1,2] by the @ symbol, is not widely understood, despite the wide acceptance of the principle of least action in mechanics discussed in Section 2.2. Here, a glossary of the factors used in its estimation is provided. Quantum mechanics requires that all material particles, however they are combined, must exist in exact quantum states. These states can only differ from other states by exact numbers of quanta of energy $h\nu$, Planck's quantum of action multiplied by frequency. Neither energy nor matter is infinitely divisible, tending to discredit infinitesimal calculus under extreme division. As a result, all motion in time and space represents a series of discrete quantum states and an infinitesimally smooth curve for motion is impossible, reminiscent of one of Zeno's paradoxes.

| | |
|---|---|
| @ | action or $Jd\phi$ or $I\omega d\phi$ where $I$ is the moment of inertia (J.s) |
| $\phi$ | indicates an angle, the ratio of circumference divided by radius expressed as radians or degrees (1 radian $\equiv$ 57.296 degrees or $2\pi \equiv$ 360 degrees |
| $d\phi$ | an infinitesimal variation in angular motion |
| $\omega$ | $\equiv d\phi/dt$ or $\Omega$, differentiating the rate of change in angular motion with time |
| $mr$ | inertia, expressed as mass modified by radius ($r$) (kg.m) |
| $mr^2$ | moment of inertia $= I$ |
| $mr^2\omega$ | angular momentum or intensity of action (J.s) |
| $mr^2\omega^2$ | energy or *vis viva* as torque ($mv^2 \equiv$ T) or twice the kinetic energy (J) |
| $mr^2\omega^3$ | power as energy per unit time $\equiv$ T$\Omega$ (J/s or W) |
| $1/r$ | curvature requiring infinite radius to achieve straight line motion or zero |
| J | energy as Joules, the work undertaken when 1 Newton acts over 1 m at constant radius from a center of force |
| $J$ | $\equiv mr^2\omega$ or angular momentum (J.s) |
| $S$ | action as $\int (T - V)dt$, the time integral of the Lagrangian $L$, the difference between kinetic and potential energy with time (J.s) |
| S | entropy or energy per unit of temperature (J/T) |

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
