# Peer review of "Applying the Action Principle of Classical Mechanics to the Thermodynamics of the Troposphere"

_2673-3161, doi:10.3390/applmech4020037_

Round 1

Reviewer 1 Report

The paper is aimed at application of action mechanics to thermodynamic analysis of atmospheric gases. It can be considered as further development of the approached proposed by one of the authors in [1] to the atmospheric gases, and by both authors in [2] – to heat engines.

The title must be corrected because the term ‘atmospheric systems’ could be defined as (i) a mixture of atmospheric gases; (ii) the same with liquids and solid particles; (iii) the system of atmospheric layers like troposphere, stratosphere, mesosphere, thermosphere, and exosphere. Since only tropospheric gases are detail considered in the paper, the better title could be “… thermodynamics of tropospheric gases’, to say.

Abstract must be structured as (1) a brief topic formulation with solved and unsolved problems and their importance; (2) the method used; (3) the main results and conclusions. Also, a definition of the operator of translational action @ must be presented. Usually the square brackets [] are reserved for the REFs; the same is for line 84. If the line 3 in the abstract presents the definition of @t, the values $s, \phi, \omega, v$ and the integration parameter must be determined. Usually the latter is $\int F(\vect{r},t)dt$ or  $\int F(\vect{r},t)dV$, V is for the volume; but not $\int F(\vect{r},t)ds/dt$. The equivalence sign for mathematical definitions is preferable. Is it a space mean translational action (since @t=mv^2)? Time mean? space&time mean? Statistical mean? It must be clarified for a wider auditory.

Some minor corrections:

Lines 39-40: the parameters $s, \omega, \phi$ must be defined. Everywhere the dot symbol is used instead of the multiplication sign.

Line 43: radial curvature of what?

Line 129: The variable J must be defined.

Line 134: why @ was substituted by S? usually S is reserved for entropy, as it is also indicated in Lines 190 and 398 S is the molar entropy.

Lines 133-136: The expression for V must be explained. What is the external force and which types of interactions between the particles are neglected?

Line 167: The variables $z_{t}, \sigma$ must be defined.

Lines 170-174: the subscripts t,r,A,B,C and the value e must be defined. Is it the same e like the ‘internal molecular energy’ in line 194? Then it must be non-dimensional and named as ‘normalized’ or ‘non-dimensional’.  Numerations (4) and (5) are unclear; are they for lines 173 and 174 only? The brackets in $e^{5/2}$ are incomplete (Line 170). The reference to ‘all textbooks on statistical mechanics’ is incorrect for they use different types of notification.

Line 193: Why ‘internal molecular energy’? Internal energy is defined as a sum of kinetic and potential energy of molecules.

Line 272: I suggest G for the Gibbs energy because g was already used for the gravitational acceleration (Line 149 and Line 354).

Line 286: It is the Gibbs energy density, not Gibbs energy.

Line 287: Here a is used as a length scale (a^3 is the volume) while in the Line 260 a was used for the absolute Gibbs energy value calculated for argon, and in the Line 295 a is the diameter of the ‘radial motion’ – radial trajectory? it is misleading.

Line 199: ‘K’ is unclear; replace it by, to say ‘at absolute zero T=0 K…’.

Lines 215 and 219: The same operator @_{t} is determined via different velocities.

Line 317: The coordinates $(r,\phi, \theta)$ are for spherical, not for radial(polar) coordinates.

Line 351: r is introduced as the mean 351 radial separation but before it was already defined in the same way (Line 184). Besides, r was also used as the polar system coordinate (Line 317) and radius of the molecule (Line 344), which is misleading.

Line 358: Is (8) is applied to the last equation only? Or two last eqs? The eqs for pressure in the lines 357 and 358 are contradictory; check powers of a and r, and the constants.

Line 398: H-G must be defined as enthalpy and Gibbs energy? Then it is easier to write down ST=H-G as eq.

Table 2: $S_{transl}$ and $S_{rotation}$ (columns 4 and 8) must be defined. Are they the same as S_{t}, S_{r} (column 11)?

Line 404: What means ‘SI data’? Data in SI units?

Line 434: The variables G, H and E must be defined before, in the lines they appeared in the text for the first time, not here. The eq. was already given before (Line 398).

Line 443: ‘radial coordinate diameter’ – just diameter, and mark this variable in Fig.1,2.

Fig.1 – 1) in the figure the letters S,E,R,C,A are used while in the list of explanations the letters s,e,r,c,a are used; 2) why spherical and ellipsoidal shapes are used for the anticyclone and cyclone; it must be explained; 3) Before the letters m,s,e,a,g,r,T were already used for other variables; they must be changed here in unused ones; 4) The choice of the same diameter for the nanoscale (molecule) and macroscale (cyclone) system with the same quantum effects must be explained; 5) The values of the variables must be explained in the caption; whether they are mean values or example ones taken from [11]; 6) Since the temperature range is T=222-288 K, Fir.1 relates to the troposphere, and the small height of both cyclone and anticyclone and their shape are unclear.

Line 470: v was already used for velocity, use another letter.

Lines 475-480: The letters r,a,e,v,g were already used for different variables, use other letters.

References:

[20] year 1999 is missed

In the list of REFs 7 papers (32%) by Kennedy I.R. among 22 others are quoted.

Only 6 REFs published during the last 5 years are quoted, and 5 of them (23%) are by Kennedy I.R.

The logics of the paper must be significantly improved. The theoretical quantum-mechanics based chapter must be clearly combiner with the macroscale cyclon-scale and wind-turbine scale approaches.

Reviewer 2 Report

Report of Review

The objective of this theoretical study is to demonstrate how quantum fields sustain the action of molecular mechanics, allowing heat to achieve work. As an illustration, the impulsive torques from quantum fields are shown to drive the reversible thermodynamics of Carnot’s heat-work engine cycle, sustain the decreasing atmospheric temperature gradient of increasing molecular entropy with altitude etc......

1. This paper is difficult to read as there are many unclear words and many are strange.

2. Also, Graphical lustrations may help better understand the studied thermodynamic cycles and transformation concepts.

3. The concept here as extended to wind-turbines seems inappropriate and infringes to the aerodynamic principles applied to the wind-turbines.

 Please address all the shortcoming and next queries in a rebuttal and show how answers are reflected in the paper text.

Queries:

4. Reformulate the title; it is not reflecting the paper subject.  

5. Check thoroughly your English language; there are several awkward sentences and fragmented sentences without coordination.  Also, there are plenty of grammatical errors and uncommon words. Please have your paper proof-edited by a professional

6. The Abstract should contain answers to the following questions: What problem was studied and why is it important? What methods were used? What are the important results? What conclusions can be drawn from the results? What is the novelty of the work and where does it go beyond previous efforts in the literature?

7. Remove equations from abstract

8. Please use  indefinite not we

9. Many symbols used here are not typical to the open literature, check carefully

10. Add the list of symbols and acronyms.

11. The paper structure is not well organized.

12.  Explain here not clear : vis viva

13. Torque is not a keyword.

14. This word is misused here line 51:   lubricated

15. Add sentences after main headings

16. Reword here: the chasm.

17. Reword here: the virial theorem

18. Reword here: heat bath

19. Correct here : enetropy

20. Explain in line 217 the term of symmetry:  more symmetry exhibited in a mechanical system.

21. Reword here line 222: its lysis.

22. Reword here line 242: entanglements  

23. Reword here line 283:quantized

24. According to what said in line 322 I do not see any of algorithms in this paper.

25. Reword here line 323: plainly

26. Correct here line 337: as we now find

27. Reword here line 338: We coined

28. Reword here line 348: erratically

29. Check table 2 some symbols and units are incorrect

30.  What you mean by the reduced Planck’s constant; Planck constant should be the same as that cited beforehand.

31.   Explain here not clear: solving an objection voiced by some.

32. Technical terms are uncommon: both windward and leeward surfaces of wind turbines.

33. In table 3 provide results based on Betz theory and compare with

34. Not clear what you have said at line 537

35. What you mean by: aftermath

36. Provide scale and ticks for fig.3, and its citation within the text.

37. Line 661 is unclear.

38. Line 677 is unclear. Revise on the quantum causation

39. Line 681 is unclear. consistent with Ockham’s razor .....

40.  Conclusion should be reformulated. Focus on the presented methodology, the main findings and practical aspects and the perspectives.

1. Check thoroughly your English language; there are several awkward sentences and fragmented sentences without coordination.  Also, there are plenty of grammatical errors and uncommon words. Please have your paper proof-edited by a professional

Round 2

Reviewer 1 Report

Now the authors have corrected the manuscript according to my remarks and the paper can be accepted.

Author Response

Thank you.

Reviewer 2 Report

Report of Review

There are significant improvements, but still there few queries not satisfied yet.

1.       It is not clear how this thermodynamic concept is applicable to wind-turbines .

2.       Remove equations from abstract

3.       Conclusion should be reformulated. Focus on the presented methodology, the main findings and practical aspects and the perspectives.

 Moderate editing of English language

Author Response

We thank Reviewer 2 for this guidance that we find helpful, further improving the manuscript.

We have responded to all these requests, adding two paragraphs before and after Section 3.4 (lines 549-575), explaining how wind turbines can also be regarded as heat engines justifying inclusion; the action mechanics approach emphasizing torques as a rate of action provide the thermodynamic viewpoint needed. Equations have been removed from the abstract but one relationship using equivalence symbols as suggested by Reviewer 1 remains. The conclusion has been reformulated using the guidance provided, including stating the main findings, practical aspects and prospects.  We draw attention to the following replacement paragraphs (lines 834-851) shown following.

  • A new method to estimate vortical energy is presented, based on the action or quantum state of molecules. This is important providing better understanding of how the Earth’s surface is heated as measured by local temperature, how wind power for wind turbines is sustained, independent of kinetic energy and how the destructive power of tropical cyclones is generated from the heat of vaporization of tropical sea water. Far more energy is stored as vortical energy than currently assumed based on laboratory measurements of heat capacity supporting internal energy. This means that more heating released from turbulence caused by collisions of air masses or released by wind farms must be considered.
  • Practical consequences from the separate analyses conducted in this article include (i) a new means to estimate the lapse rate in the atmosphere that can inform climate models. (ii) a method to model heat absorption in the troposphere by greenhouse gases for recycling (Fig. 1) by driving vortical friction at the surface boundary layer. (iii) a new means to estimate turbulent heat production downwind of wind turbines, suitable for future field studies (iv) a better understanding of the heat-work cycle involved in tropical cyclones, with heat radiated by condensation of water at the eyewall powering the vortical action of the cyclone. All of these findings offer opportunities for new means of gathering information by testing technologies.

  Other minor changes have also been made that can be seen in the track changes version, or by comparing version 3 with 2.  These all emphasize how the principle of least action can be considered as having thermodynamic application too. Thank you for your assistance. 

Sincerely,

Ivan R. Kennedy AM FRACI

Professor Emeritus in Agricultural & Environmental Chemistry.

University of Sydney